# The Supervised Information Bottleneck

**DOI:** 10.3390/e27050452

**Published:** 2025-04-22

**Authors:** Nir Z. Weingarten, Zohar Yakhini, Moshe Butman, Ronit Bustin

**Affiliations:** 1Department of Computer Science, Reichman University, Herzliya 4610101, Israel; zohar.yakhini@gmail.com (Z.Y.); moshebu@colman.ac.il (M.B.); 2Toga Networks, a Huawei Company, Tel Aviv 4524075, Israel; ronit.bustin@gmail.com

**Keywords:** information theory, information bottleneck, machine learning

## Abstract

The Information Bottleneck (IB) framework offers a theoretically optimal approach to data modeling, although it is often intractable. Recent efforts have optimized supervised deep neural networks (DNNs) using a variational upper bound on the IB objective, leading to enhanced robustness to adversarial attacks. In these studies, supervision assumes a dual role: sometimes as a presumably constant and observed random variable and at other times as its variational approximation. This work proposes an extension to the IB framework and, consequent to the derivation of its variational bound, that resolves this duality. Applying the resulting bound as an objective for supervised DNNs induces empirical improvements and provides an information-theoretic motivation for decoder regularization.

## 1. Introduction

The Variational Information Bottleneck (VIB) [1] adapts the theoretically optimal, yet mostly intractable, Information Bottleneck (IB) [2] to supervised DNNs. However, the IB is a method for unsupervised learning and requires knowledge of the underlying joint distribution p(x,y) [3]. This requirement is relaxed in the original VIB derivation, resulting in a duality of the usage of the downstream RV *Y*, which is treated both as an observed RV when sampled from the training data, and as a variational approximation when optimized. This work proposes a new adaptation of the IB and VIB frameworks for supervised tasks, and consequently, an information-theoretic motivation for decoder regularization.

We begin by laying down what IB is and how it can be adapted as an objective for DNNs—classic information theory provides rate-distortion [4] for optimal data compression. However, rate distortion regards all information as equal, not taking into account which information is more relevant to a specified downstream task without constructing tailored distortion functions. The Information Bottleneck (IB) [2] resolves this limitation by defining mutual information (MI) between the learned representation and a designated downstream random variable (RV) as a universal distortion function. Yet, learning representations using the IB method is possible given discrete distributions and some continuous ones, but not in the general case [5]. Moreover, MI is either difficult or impossible to optimize when considering deterministic models, such as DNNs [6,7]. Nonetheless, the promise of the IB remains alluring, and recent efforts have utilized VAE [8] inspired variational methods to approximate upper bounds on the IB objective, allowing for its utilization as a loss function for DNNs, where the underlying distributions are both continuous and unknown [1,9,10,11]. These approaches learn representations in supervised settings without knowledge of the underlying distribution p(x,y), utilizing the learned variational conditional p(y|x) to approximate MI. In contrast, non-variational IB methods learn representations in unsupervised settings, where the stochastic process underlying the observed data is known [2,5,12]. Nonetheless, when deriving the variational IB objectives, previous research considered the learned representation as the only optimized RV when, in practice, a variational classifier is also optimized.

This work proposes an extension of the IB and variational IB objectives by setting the downstream RV as a parameterized model in the problem definition. An empirical comparison of models trained with the proposed objective, and identical models trained using previous IB adaptations, demonstrates improved performance in several cases across various challenging tasks and modalities. Finally, interpreting our findings in the context of previous work in the field leads us to propose a novel information-theoretic interpretation of overfitting in supervised DNNs.

The reader is encouraged to refer to the preliminaries provided in Appendix A before proceeding.

## 2. Materials and Methods

### 2.1. Related Work

**Deterministic Information Bottleneck** Classic information theory offers rate distortion [4] to mitigate signal loss during compression. A source *X* is compressed to an encoding *Z*, such that maximal compression is achieved while keeping the encoding quality above a certain threshold. Encoding quality is measured by a task-specific distortion function: $d:X\times Z\mapsto R^{+}$. Rate distortion suggests a mapping that minimizes the rate of bits to the source sample, measured by I(X;Z), that adheres to a chosen allowed expected distortion D≥0. The Information Bottleneck (IB) [2] extends rate distortion by replacing the tailored distortion functions with MI over a target distribution. Let *Y* be the target signal for some specific downstream task, such that the joint distribution p(x,y) is known, and define the distortion function as MI between *Z* and *Y*. The IB is the solution to the optimization problem $Z:\underset{p(z|x)}{min}I\left(X;Z\right)$ subject to I(Z;Y)≥D, which can be optimized by minimizing the IB objective $L_{IB}=\beta I\left(X;Z\right)-I\left(Z;Y\right)$ over p(z|x). The solution to this objective is a function of the Lagrange multiplier β, and is a theoretical limit for representation quality, given mutual information as an accepted metric, as elaborated in more detail in Appendix B. The IB is, in fact, an unsupervised soft clustering problem in which each data point *x* is assigned a probability *z* of belonging to different clusters, given the joint distribution of the input and target tasks p(x,y) [3]. Chechik et al. [5] showed that computing the IB for continuous distributions is hard in the general case and provided a method for optimizing the IB objective in the case where X,Y are jointly Gaussian and known. Painsky and Tishby [12] offered a limited linear approximation of the IB for any distribution by extracting the joint Gaussian element of given distributions. Saxe et al. [6] considered the application of the IB objective as a loss function for DNNs and concluded that computing mutual information in deterministic DNNs is problematic as the entropy term H(Z|X) for a continuous *Z* is infinite. Amjad and Geiger [7] extended this observation and pointed out that for a discrete *Z*, MI becomes a piecewise constant function of its parameters, making gradient descent limited and difficult.

Considering the supervised problem, Geiger and Fischer [13] suggested regarding the classification output as an additional random variable, leading to an extended underlying Markov chain: $Y↔X↔Z↔\tilde{Y}$. A similar approach has also been suggested by Piran et al. [14], where a dual IB formulation was proposed that, while still considering the minimization of I(X;Z), replaces the constraints with one that takes into account $\tilde{Y}$. The approach suggested here follows these ideas but adds the additional objective of reducing overfitting during the classification step.

**Variational Information Bottleneck** Alemi et al. [1] introduced the Variational Information Bottleneck (VIB), a variational approximation for an upper bound to the IB objective for DNN optimization. Bounds for I(X,Z) and I(Z,Y) are derived from the non-negativity of KL divergence and are used to form an upper bound for the IB objective. Variational approximations are then used to replace intractable distributions in the upper bound. Using the *reparameterization trick* [8], a discrete empirical estimation of the variational upper bound is used as a loss function for classifier DNN optimization, resulting in an objective that is equivalent to the β-autoencoder loss [15]. VIB was evaluated over image classification tasks and displayed substantial improvements in robustness to adversarial attacks while inflicting a slight reduction in test set accuracy when compared to equivalent deterministic MLE models. The improved robustness is attributed to an improvement in representation quality and, subsequently, better generalization. Achille and Soatto [11] extended VIB with a total correlation term, designed to increase latent disentanglement.

Kolchinsky et al. [10] and Cheng et al. [16] derived variational upper bounds for the IB objective that match the VIB formulation for I(Z;Y), while leveraging different MI estimators [16,17] to bound I(X;Z). These approaches demonstrated improved performance over VIB on several tasks, although they can be challenging to scale to high-dimensional settings [10,18]. In a complementary direction, Yu et al. [19] proposed a non-variational method for estimating I(X;Z), showing improvements over VIB on several low-dimensional datasets.

Fischer [9] proposed an IB-based loss function called the Conditional Entropy Bottleneck (CEB), in which the conditional mutual information of *X* and *Z* given *Y* is minimized instead of the unconditional mutual information. The CEB loss, $L_{CEB}=\underset{Z}{min}I\left(X;Z|Y\right)-\gamma I\left(Y;Z\right)$, is designed to minimize all information in *Z* that is not relevant to the downstream task *Y*, by conditioning over *Y*. CEB is equivalent to IB for γ=β−1, following the chain rule of mutual information [20] and the IB Markov chain, as established in Appendix B. However, its variational approximation, VCEB, differs from VIB in how the marginal is approximated. Geiger and Fischer [13] showed that VCEB is a tighter variational approximation for IB under certain conditions but not in the general case. Empirical studies carried out in [9] demonstrated that VCEB improves accuracy and robustness to targeted PGD attacks [21] for F-MNist [22], and increased robustness to untargeted PGD on Cifar-10 [23], compared to VIB and deterministic MLE models. Later work [24] involved an extensive experimental investigation that further substantiated the gains of VCEB, demonstrating that models trained with VCEB achieved improved robustness over deterministic MLE models for the targeted PGD attack [21] on ImageNet [25], and improved classification accuracy on the ImageNet-A and ImageNet-C datasets, two flavors of ImageNet that assess model performance on challenging edge cases and robustness to common corruptions.

The work carried out in [24] offers an in-depth comparison of VCEB to deterministic MLE models, training 80 VCEB and six MLE ImageNet classifiers from the ground up while exploring varying architectures, hyperparameters, pretraining strategies, and optimization techniques to compare the best possible VCEB model to its deterministic MLE counterpart. In contrast, the experiments in [1,9] and in the present study focus on comparing the performance of otherwise identical models using different objective functions—deterministic MLE, VIB, and VCEB, as well as SVIB in the current study.

**Information-Theoretic Regularization** Label smoothing [26] and entropy regularization [27] regularize classifier DNNs by increasing classifier entropy, either by inserting a scaled conditional entropy term to the objective, or by smoothing the training labels. Applying either method improved test accuracy and model calibration on various challenging tasks.

Alemi et al. [28] extended the information plane [2] to VAE [8] settings, measuring distortion as MI between input and reconstructed images and rate as the KL divergence between variational representation and marginal. The limits of representation quality in VAEs are looser than the theoretical IB limits and heavily depend on the chosen variational families of the marginal and decoder distributions. The closer the families are to the true distributions, the tighter the gap to the theoretical IB limit for representation quality. Alemi et al. [28] also showed that given a strong enough decoder, the ELBO loss is prone to produce low-quality representations, as the ELBO KL regularization term might induce completely uninformative representations that are then overfitted by the powerful decoder, as elaborated in detail in Appendix B.

In the current study, a conditional entropy term [27] emerges during the derivation of our proposed adaptation of the IB objective, providing a possible remedy to the discrepancies in the ELBO loss described in [28], and subsequently VIB and VCEB loss functions.

### 2.2. From VIB to SVIB

**Problem Definition** As elaborated in Section 2.1, the IB objective, $L_{IB}=\beta I\left(X;Z\right)-I\left(Z;Y\right)$, is computed over the joint distribution p(x,y,z). When p(x,y) is given, this expression is optimized over the distribution p(z|x,y), as proposed by Tishby et al. [2]:(1)$$\begin{array}{cc} \; & \min_{p(z|x,y)}I\left(Z;X\right)\\ \; & s.t.\;I\left(Z;Y\right)\geq D_{1}. \end{array}$$
However, adapting IB to supervised tasks admits the learned classifier as a new RV to the optimization problem [13,14]. Thus, we consider the extended Markov chain $Y↔X↔Z↔\tilde{Y}$ for the supervised IB, distinguishing between the true unknown RV *Y*, and the learned classifier $\tilde{Y}$. We follow this approach and also assume that $\tilde{Y}$ and *Y* share the same support. The IB framework connects the underlying joint distribution of the input and objective data, p(x,y), with a learned representation *Z*. We claim that when applying IB to supervised tasks, one must also consider the connection to the classifier defined by the output RV $\tilde{Y}$. Thus, we also want to consider the joint distribution over the pair $Z,\tilde{Y}$ during optimization. Following the IB method logic, we seek a $\tilde{Y}$ that will minimize mutual information with *Z* while keeping below a defined distortion metric with the true *Y*. That is, we seek a second bottleneck that minimizes the passage of information between *Z* and $\tilde{Y}$ so as to limit it to the minimum required to ensure that $\tilde{Y}$ is similar enough to *Y*, given the transition through both *X* and *Z*. Since, in this case, we are optimizing over the joint conditional distribution $p(z,\tilde{y}|x,y)$, instead of the conditional $p(\tilde{y}|z,x,y)$, this problem is not simply an IB problem over the Markov chain $Y↔Z↔\tilde{Y}$. Additionally, contrary to the standard IB, *X* plays a significant role, controlling the distribution of *Z*, and the entire chain of four random variables must be taken into consideration. We thus define a second bottleneck for the true distribution p(x,y) and modeled distribution $c\left(\tilde{y}|z\right)p\left(z|x,y\right)$. We choose KL divergence as a distortion metric, as we assume *Y* and $\tilde{Y}$ share the same support. For some positive scalar $D_{2}$ we have:(2)$$\begin{array}{cc} \; & \min_{c\left(\tilde{y}|z\right)p\left(z|x,y\right)}I\left(Z;\tilde{Y}\right)\\ \; & s.t.\;D_{KL}\left(p\left(y=\tilde{y}|z,x\right)||c\left(\tilde{y}|z\right)p\left(z|x\right)\right)\leq D_{2}. \end{array}$$
Combining the two bottlenecks results in a new optimization problem, which we denote the Supervised Information Bottleneck (SIB), which minimizes the following objective:(3)$$\begin{array}{cc} \; & L_{SIB}\equiv \beta I\left(X;Z\right)-I\left(Z;Y\right)+\lambda I\left(Z;\tilde{Y}\right)+D_{KL}\left(p\left(y=\tilde{y}|z,x\right)||c\left(\tilde{y}|z\right)p\left(z|x\right)\right). \end{array}$$

**Optimization Objective** We proceed to derive a tractable variational upper bound for $L_{SIB}$, which we can use as an objective function for classifier DNNs. We begin by deriving the first bottleneck (Equation 1) as done in VIB [1], and proceed to derive the second (Equation 2).

Consider I(Z;X):(4)$$\begin{array}{c} I\left(Z;X\right)=\int \int p\left(x,z\right)\log\left(p(z|x)\right)\;dx\;dz-\int p\left(z\right)\log\left(p(z)\right)\;dz. \end{array}$$

For any probability distribution *r*, we have that $D_{KL}\left(p(z)||r(z)\right)\geq 0$. It follows that:(5)$$\int p\left(z\right)\log\left(p(z)\right)\;dz\geq \int p\left(z\right)\log\left(r(z)\right)\;dz.$$

So, by Equation (Equation 5):(6)$$I\left(Z;X\right)\leq \int \int p\left(x\right)p\left(z|x\right)\log\left(\frac{p(z|x)}{r(z)}\right)\;dx\;dz,$$
Consider I(Z;Y):   By the Barber–Agakov inequality [29], we have that for any probability distribution *c*:(7)$$\begin{array}{c} I\left(Z;Y\right)\geq \int \int p\left(y,z\right)\log\left(c(y|z)\right)\;dy\;dz-\int p\left(y\right)\log\left(p(y)\right)\;dy. \end{array}$$

Note that Equations (Equation 6) and (Equation 7) hold for any distribution *r* over the support of *Z*, and for any conditional distribution c(·|z) whose support equals the support of *Y* for every given value *z* in support of *Z*. We link the two bottlenecks by choosing *c* to be $\tilde{Y}|Z=z∼c\left(\cdot |z\right)$, meaning the variational classifier distribution. This connection is implicit in [1], where $\tilde{Y}$ is not formally defined. We now move on to the second bottleneck.

Consider $I(Z;\tilde{Y})$:(8)$$\begin{array}{cc} I(Z;\tilde{Y}) & =H\left(\tilde{Y}\right)-H\left(\tilde{Y}|Z\right). \end{array}$$
Choosing a discrete random variable for $\tilde{Y}$, as in labeled classification, we have $H\left(\tilde{Y}\right)\leq \log\left\|\tilde{Y}\right\|$. Otherwise, choosing a continuous RV with finite support [a,b], we have that $H\left(\tilde{Y}\right)\leq \log\left(b-a\right)$. In both cases, $I(Z;\tilde{Y})$ is bounded from above by some constant J=log(b−a), or $J=\log\|\tilde{Y}\|$, and the negative conditional entropy term $-H(\tilde{Y}|Z)$:(9)$$\begin{array}{cc} I(Z;\tilde{Y}) & \leq J-H\left(\tilde{Y}|Z\right)=J+\int \int p\left(\tilde{y},z\right)\log\left(c\left(\tilde{y}|z\right)\right)\;d\tilde{y}\;dz. \end{array}$$
Consider $D_{KL}\left(p\left(y=\tilde{y}|z,x\right)||c\left(\tilde{y}|z\right)p\left(z|x\right)\right)$:(10)$$\begin{array}{cc} D_{KL}\left(p\left(y=\tilde{y}|z,x\right)||c\left(\tilde{y}|z\right)p\left(z|x\right)\right)=\\ \int \int \int p\left(y,z,x\right)\log\left(p(y|z,x)\right)\;dy\;dx\;dz- & \int \int \int p\left(y,z,x\right)\log\left(c(y|z,x)\right)\;dy\;dx\;dz. \end{array}$$
Applying the Markov chain $Y↔X↔Z↔\tilde{Y}$, and total probability, we get:(11)$$\begin{array}{c} D_{KL}\left(p\left(y=\tilde{y}|z,x\right)||c\left(\tilde{y}|z\right)p\left(z|x\right)\right)=\\ \;\;\;\;\;\;\;\;\;\;\;\;\;\;\;\;\;\;\;\;\;\;\;\int \int p\left(y,x\right)\log\left(p(y|x)\right)\;dy\;dx-\int \int p\left(y,z\right)\log\left(c(y|z)\right)\;dy\;dz. \end{array}$$

Finally, we attain an upper bound for $L_{SIB}$ by combining Equations (Equation 6), (Equation 7), (Equation 9), and (Equation 11):(12)$$\begin{array}{cc} L_{SIB}\leq & \beta \int \int p\left(x\right)p\left(z|x\right)\log\left(\frac{p(z|x)}{r(z)}\right)\;dx\;dz-2\int \int p\left(y,z\right)\log\left(c(y|z)\right)\;dy\;dz\\ + & \lambda \int \int c\left(y|z\right)p\left(z\right)\log\left(c(y|z)\right)\;dy\;dz+\int \int p\left(y,x\right)\log\left(p(y|x)\right)\;dy\;dx\\ + & \int p\left(y\right)\log\left(p(y)\right)\;dy+\lambda J. \end{array}$$Note that p(x,y) and *J* are constants, so the last three terms in Equation (Equation 12) can be ignored in the course of optimization.

**Variational Approximation and Empirical Estimation** We further develop the upper bound in Equation (Equation 12) using the IB Markov chain $Y↔X↔Z↔\tilde{Y}$ and total probability, and define tractable variational distributions to replace intractable ones. Let e(z|x) a variational encoder approximating the conditional p(z|x), let r(z) be a variational approximation for the marginal, and let c(y|z) be a variational classifier approximating p(y|z). We define the variational approximation $L_{SVIB}$:(13)$$\begin{array}{cc} L_{SVIB}\equiv & \beta \int \int p\left(x\right)e\left(z|x\right)\log\left(\frac{e(z|x)}{r(z)}\right)\;dx\;dz\\ - & 2\int \int \int p\left(x\right)p\left(y|x\right)e\left(z|x\right)\log\left(c(y|z)\right)\;dx\;dy\;dz\\ + & \lambda \int \int \int p\left(x\right)e\left(z|x\right)c\left(y|z\right)\log\left(c(y|z)\right)\;dx\;dy\;dz. \end{array}$$

As is common in the VIB and VAE literature, we choose a standard Gaussian for the variational marginal r(z), a spherical Gaussian for the variational encoder e(z|x), and a categorical distribution for the variational classifier c(y|z). We use DNNs to model these distributions as follows: Let $e_{\phi }\left(z|x\right)∼N\left(\mu ,\Sigma \right)$ be a stochastic DNN encoder with parameters ϕ, and a final layer of dimension 2K, such that for each forward pass, the first *K* entries are used to encode μ, and the last *K* entries to encode a diagonal Σ, after a soft-plus transformation. Let $C_{\gamma }$ be a discrete classifier neural net parameterized by γ, such that $C_{\gamma }\left(y|z\right)∼Categorical$. r(z) is constant and unparameterized. We use Monte Carlo sampling over some discrete dataset S to empirically estimate $L_{SVIB}$. The true and possibly continuous distribution p(x,y)=p(y|x)p(x) can be sampled from S. Distributions featuring *Z* are samples from the stochastic encoder using the *reparameterization trick* [8], such that for each $x_{n}\in S$ we generate a sample ${\hat{z}}_{n}$. Finally, we use the variational classifier to attain instances ${\tilde{y}}_{n}$, given an instance ${\hat{z}}_{n}$.(14)$${\hat{L}}_{SVIB}\equiv \frac{1}{N}\sum_{n=1}^{N}\left[\beta D_{KL}\left(e_{\phi }\left(z|x_{n}\right)||r\left(z\right)\right)-\log\left(C_{\gamma }\left(y_{n}|{\hat{z}}_{n}\right)\right)+\lambda \log\left(C_{\gamma }\left({\tilde{y}}_{n}|{\hat{z}}_{n}\right)\right).\right]$$

The only difference between VIB and SVIB lies in the additional conditional entropy term $\lambda \log\left(C_{\gamma }\left({\tilde{y}}_{n}|{\hat{z}}_{n}\right)\right)$. This term can be computed at each iteration using the existing logits, preserving the overall asymptotic complexity of $O\left(N\cdot \|Y\|\cdot \|X\|\right)$. Detailed runtime performance measurements are provided in Appendix C.

**Motivation** Tishby et al. [2] proposed that representations are optimal if they contain just enough information for a required downstream task, and that the Information Bottleneck is a method to obtain such representations. However, in the supervised case, an additional information processing stage is added, where representations are decoded by a learned decoder (here, decoder in the general sense, including classifiers and other decoders) in a joint training process. As mentioned in Section 2.1, Alemi et al. [28] observed that the ELBO loss function [8] may learn uninformative representations even when strong KL regularization is imposed, since an overpowerful decoder can overfit the learned embeddings. This observation holds for all VIB loss functions [1,9,16], as VIB is equivalent to the ELBO loss, as shown in [1]. Our proposed extension to the IB and VIB frameworks asks to resolve this conflict. By appending an additional bottleneck between representation *Z* and learned classifier $\tilde{Y}$, we learn a classifier that holds the minimum information about the representation that is required to meet a designated distortion target over the true downstream RV. Extending the work in [28], we propose to define a decoder $\tilde{Y}$ as overfitting if a substantial amount of its information about *Z* lacks relevance about *Y*. The conditional MI $I(Z;\tilde{Y}|Y)$ measures the amount of information *Z* and $\tilde{Y}$ share, which is uninformative about *Y*. Hence, we have that $\tilde{Y}$ overfits *Z* if:(15)$$\begin{array}{cc} I(Z;\tilde{Y})\gg & I\left(Z;\tilde{Y}\right)-I\left(Z;\tilde{Y}|Y\right)\\ H(\tilde{Y}|Y)\gg & H(\tilde{Y}|Z), \end{array}$$

Where the last line follows from the SIB Markov chain.

By deriving the second bottleneck, $L_{SVIB}$ introduces a modulated conditional entropy term to the loss function, $-\lambda H(\tilde{Y}|Z)$, inducing an increase in the right-hand side of Equation (Equation 15). At the same time, we expect that the left-hand side conditional entropy will be reduced by the power of the cross entropy term. Applying these two forces together prevents decoders from overfitting embeddings, as illustrated in Figure 1.

## 3. Results

We follow the experimental setup proposed by Alemi et al. [1], extending it to NLP tasks as well. We trained image classification models on the ImageNet 2012 dataset [25], and text classification models on the IMDB sentiment analysis dataset [30]. For each dataset, we compared a competitive deterministic MLE model with VIB models trained over eight different β values ranging from $10^{-4}$ to 0.5, VCEB models trained with ρ values ranging from 1 to 7, and SVIB models trained with different combinations of β and λ values. Each model was trained and evaluated five times per setting. As in [1], all models were trained over a frozen encoder of the deterministic model instead of training models from the ground up, enabling extensive testing while meeting resource constraints. Models were evaluated for test set accuracy and robustness to various adversarial attacks, showing consistent performance. For image classification, we employed the untargeted Fast Gradient Sign (FGS) attack [31], as well as the targeted CW $L_{2}$ attack [32,33]. For text classification, we used the untargeted Deep Word Bug attack [34,35] as well as the untargeted PWWS attack [36]. The empirical results presented in Figure 2 confirm that while VIB, VCEB, and SVIB models mostly decrease test set accuracy compared to the deterministic MLE model, they significantly improve robustness to the applied adversarial attacks. SVIB consistently attains higher test set accuracy over VIB and VCEB, and in one case over the deterministic model as well, while demonstrating improved or on-par robustness in all attacks, apart from the targeted CW attack. A comparison of the best MLE, VIB, VCEB, and SVIB models further substantiates these findings, with statistical significance confirmed by a *p*-value of less than 0.05 on a Wilcoxon rank sum test.

As in [1,9], the experiments performed in the current study compare identical models that differ only in their objective function, ensuring that any performance differences arise solely from these variations. Rather than benchmarking the best possible performance, these experiments serve to validate the proposed information-theoretic approach. This method enabled us to perform a direct comparison of four objective functions across high-dimensional tasks in two modalities while systematically exploring the entire range of useful β, λ, and ρ values, with five runs per setting. We leave for future work the task of benchmarking SVIB for state-of-the-art performance. As carried out in [24], this would involve training ImageNet classifiers from the ground up without a pre-trained encoder, experimenting with larger model architectures, using Gaussian-mixture-models for *Z*, training for more epochs and incorporating training techniques such as AutoAugment [37], $L_{2}$ weight decay, and annealing strategies for β and λ.

Elaboration on the experimental setup, detailed results, and further insights from the experiments are available in Appendix C. The code for reconstructing the experiments is provided in the following Github repository: github.com/nirweingarten/svib (accessed on 16 April 2025).

### 3.1. Image Classification

A pre-trained inceptionV3 [26] base model was used and achieved a 77.21% accuracy on the ImageNet 2012 validation set. Image classification evaluation results are shown in Figure 2, and examples of successful attacks are shown in Figure 3 and Figure 4. The ImageNet 2012 validation set was used for evaluation as the test set for ImageNet is unavailable. InceptionV3 yields a slightly worse single shot accuracy than inceptionV2 (80.4%) when run in a single model and single crop setting; however, we have used InceptionV3 over V2 for simplicity. Each model was trained for 100 epochs. The entire validation set was used to measure accuracy and robustness to FGS attacks, while only 1% of it was used for CW attacks, as they are computationally expensive. Complete results are available in Section C.1. Examples of successful attacks are shown in Figure 3 and Figure 4. t-SNE [38] visualization of the latent space of each model is presented in Figure 5.

### 3.2. Text Classification

A fine-tuned BERT uncased [39] base model was used and achieved a 93.0% accuracy on the IMDB sentiment analysis test set. Text classification evaluation results are shown in Figure 2. Each model was trained for 150 epochs. The entire test set was used to measure accuracy, while only the first 200 entries in the test set were used for adversarial attacks, as they are computationally expensive. Complete results are available in Section C.1. Examples of successful attacks are shown in Table 1 and Table 2.

## 4. Discussion

The IB is a special case of rate distortion, and was initially designed to optimize compressed representations. Applying the IB objective for supervised tasks results in the optimization of a classifier distribution as well, and requires a reformulation of the initial problem to include both representation and classification. We propose the Supervised IB (SIB), an extension to the original IB that considers the classifier distribution as well, and adds an additional bottleneck to mitigate information flow between representations and the classifier. We derive SVIB, a tractable variational approximation for SIB, and show that it induces empirical gains in terms of classification accuracy and robustness to several adversarial attacks over high dimensional tasks of different modalities, with high statistical significance. We apply previous information-theoretic frameworks for deep learning [26,27,28] to interpret our findings and propose a definition for decoder overfitting, and a new motivation for conditional entropy regularization. While other advancements have been achieved in recent years, [1,9,10,11], none propose a reformulation for IB, as is required in our opinion.

This study opens many opportunities for further research: Applying SVIB in self-supervised learning, and in particular, measuring whether representations learned with SVIB capture better semantics than representations learned with non-IB inspired loss functions; An in-depth empirical study that includes training an encoder from the ground up with SVIB, exploring different architectures, using a full covariance matrix and Gaussian-mixture-model for *Z*, applying different training techniques such as AutoAugment, β and λ annealing, and $L_{2}$ weight decay, as well as testing against PGD [21] and AutoAttack [40]; Combining SVIB with VCEB is also left for future work.

### Limitations

SVIB falls short of VIB and VCEB in robustness to the targeted CW attack, introduces an additional hyperparameter, and requires a slightly longer training time than VIB due to the additional entropy computation.

## Figures and Tables

**Figure 1 entropy-27-00452-f001:**
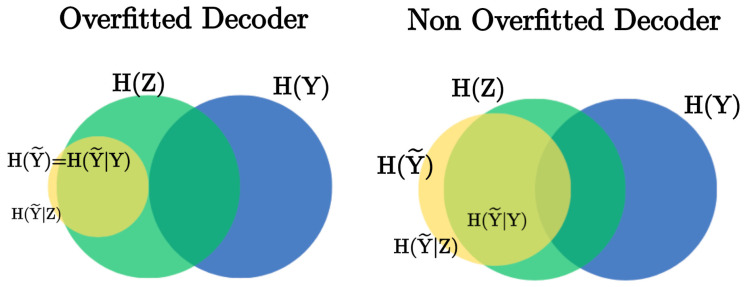
Venn diagrams illustrating decoder overfitting. The left diagram depicts an overfitted decoder where $\tilde{Y}$ holds no information about *Y*, and $H\left(\tilde{Y}|Y\right)\gg H\left(\tilde{Y}|Z\right)$. The right diagram depicts a regularized decoder where $H(\tilde{Y}|Y)$ is not much greater than $H(\tilde{Y}|Z)$.

**Figure 2 entropy-27-00452-f002:**
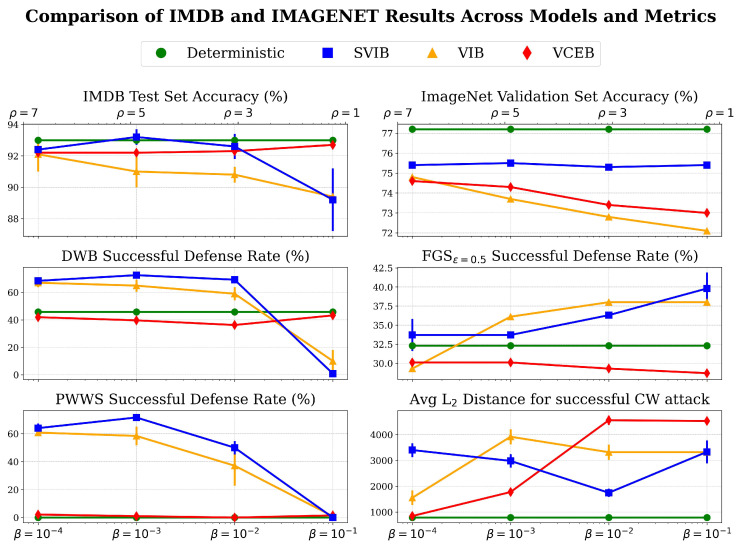
Performance comparison across models and metrics for IMDB and ImageNet. **Higher is better ↑ in all plots**. Analyzing accuracy and robustness against adversarial attacks for deterministic MLE, SVIB, VIB, and VCEB models under varying β and ρ values, averaged over five runs with standard deviation. Left column features IMDB tasks, right column features ImageNet tasks. Upper row shows accuracy over test set, and bottom rows depict robustness under various adversarial attacks, presented as the rate of deflected attacks or as the average $L_{2}$ distance required for a successful CW attack. Results show that SVIB consistently attains higher test set accuracy and higher or on-par robustness in all attacks apart from the targeted CW attack. ρ values apply to VCEB models, while β values apply to SVIB and VIB models. SVIB results are presented for λ=1 in IMDB and λ=2 in ImageNet. For all experimental results please see Section C.1.

**Figure 3 entropy-27-00452-f003:**
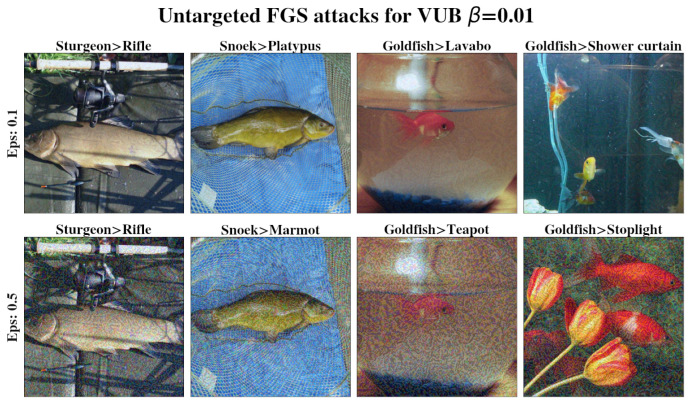
Successful untargeted FGS attack examples. Images are perturbations of previously successfully classified instances from the ImageNet validation set. Perturbation magnitude is determined by the parameter ϵ shown on the left—the higher, the more perturbed. Original and wrongly assigned labels are listed at the top of each image. Notice the deterioration of image quality as ϵ increases.

**Figure 4 entropy-27-00452-f004:**
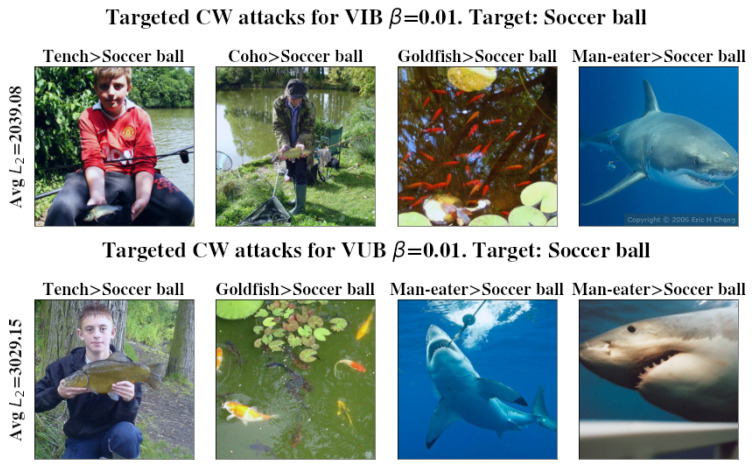
Successfully targeted CW attack examples. Images are perturbations of previously successfully classified instances from the ImageNet validation set. The target label is ‘Soccer ball’. Average $L_{2}$ distance required for a successful attack is shown on the left. The higher the required $L_{2}$ distance, the greater the visible change required to fool the model. Original and wrongly assigned labels are listed at the top of each image. Mind the difference in noticeable change as compared to the FGS perturbations presented in Figure 3.

**Figure 5 entropy-27-00452-f005:**
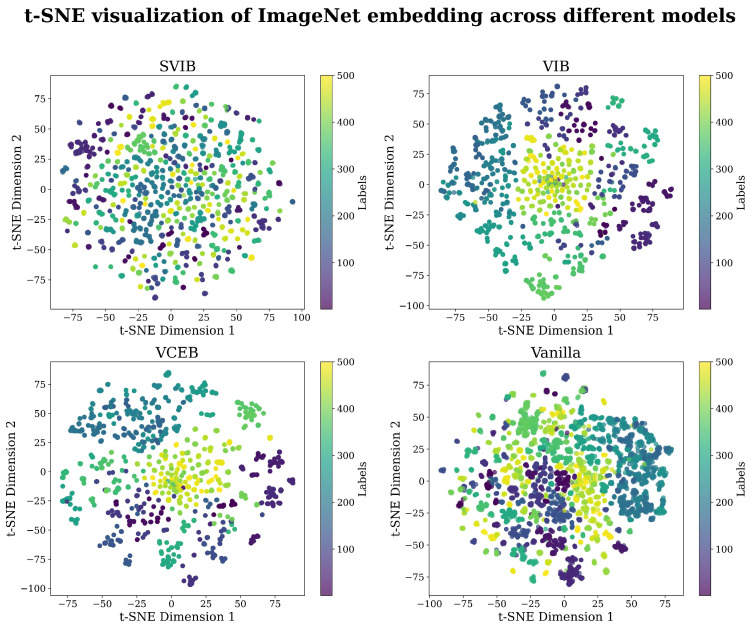
ImageNet embeddings of the different models casted to 2D using the t-SNE algorithm [38]. 5000 datapoints of the first 500 ImageNet labels. The VIB and VCEB castings share similar traits of well-separated clusters, while the deterministic MLE casting shows some clustering that seems less formed and unseparated. The SVIB casting shows very little clustering and features the most dispersed distribution. The visualization suggests that the conditional entropy term in SVIB has negated the clustering effect of the ELBO loss and induced a more uniform representation.

**Table 1 entropy-27-00452-t001:** Example of a successful PWWS attack on a vanilla Bert model, fine-tuned over the IMDB dataset. The original label is ‘positive sentiment’. The substituted word, marked in italic font, changed the classification to ‘negative sentiment’. SVIB and VIB classifiers are far less susceptible to these perturbations, as shown in Figure 2.

Original Text
the acting, costumes, music, cinematography and sound are all *astounding* given the production’s austere locales.
**Perturbed Text**
the acting, costumes, music, cinematography and sound are all *dumbfounding* given the production’s austere locales.

**Table 2 entropy-27-00452-t002:** Example of a successful Deep Word Bug attack on a vanilla Bert model, fine-tuned over the IMDB dataset. The original label is ‘positive sentiment’. Perturbations, marked in italic font, change the classification to ‘negative sentiment’. SVIB and VIB classifiers are far less susceptible to these perturbations, as shown in Figure 2.

Original Text
*great* historical movie, will not allow a viewer to leave once you begin to watch. View is presented differently than displayed by most school books on this *subject*. My only fault for this movie is it was photographed in black and white; wished it had been in color *…* wow !
**Perturbed Text**
*gnreat* historical movie, will not allow a viewer to leave once you begin to watch. View is presented differently than displayed by most school books on this *sSbject*. My only fault for this movie is it was photographed in black and white; wished it had been in color *…* wow !

## Data Availability

This paper uses ImageNet 2012 [25] and the IMDB sentiment analysis [30] datasets which are both publicly available. The paper’s code repository features code that downloads, preprocesses and uses these datasets.

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
