# Peer review of "The Supervised Information Bottleneck"

_entropy, 2025, doi:10.3390/e27050452_

Round 1
Reviewer 1 Report
Comments and Suggestions for Authors
This paper revisits the variational information bottleneck and extends it to a supervised variational information bottleneck. The experiment on ImageNet and text classification shows that SIB achieves better results than VIB. Overall, the paper looks very good, and the theoretical parts are solid. I only have some minor suggestions as follows:
(1) Could you please provide more experiments on PGD and AutoAttack?
(2) Could you provide some computational complex comparison with VIB?
(3) Add some limitations of the SIB.
(4) Some very related works for deep deterministic information bottleneck [1] and Nonlinear information bottleneck [2]. Please add them in your introduction part.
[1] Kolchinsky, Artemy, Brendan D. Tracey, and David H. Wolpert. "Nonlinear information bottleneck." Entropy 21.12 (2019): 1181.
[2] Yu, Xi, Shujian Yu, and José C. Príncipe. "Deep deterministic information bottleneck with matrix-based entropy functional." ICASSP 2021-2021 IEEE International Conference on Acoustics, Speech and Signal Processing (ICASSP). IEEE, 2021.
Author Response
Dear Reviewer,
Thank you for your detailed and valuable feedback. We're pleased that you found the theoretical foundations of the work sound and the derivations solid. Based on your comments, we've made the following revisions to the manuscript:
Computational Complexity Comparison to VIB
We've added a complexity analysis to the Materials and Methods section and included empirical runtime comparisons in Appendix C. Both SVIB and VIB have the same asymptotic complexity of O(N * |X| * |Y|), but the additional conditional entropy term in SVIB introduces a slight increase in actual training time.
Limitations of SIB
We've added a paragraph in the Discussion section outlining SVIB’s limitations. These include sensitivity to the CW attack, the introduction of an additional hyperparameter, and the moderately increased training time.
Related Works [1], [2]
Thank you for pointing out these important contributions. We've now cited and discussed them in the Introduction, Related Work and Discussion sections. Their approach to MI estimation for $I(X;Z)$ can be combined with SVIB, and we plan future work to explore these avenues, together with other MI estimators such as CLUB [3].
PGD and AutoAttack Experiments
We agree that PGD and AutoAttack evaluations would enhance the empirical robustness of our results. Unfortunately, we've substantially overused our compute budget for this study. Our setup already extends the original VIB benchmark [4] with additional modalities, baselines (VIB, SVIB, and VCEB), and attacks, while sweeping across the entire range of relevant \beta and \lambda values with 5 independent runs per setting for statistical reliability. Given this extent, and our already depleted budget, it would be impossible for us to add additional attacks in the current work.
Nevertheless, we are actively preparing a follow-up study with extended compute resources and research assistance. This will include training on ImageNet from scratch, exploring richer posteriors (e.g., full-covariance, GMM), and adversarial evaluations using PGD, AutoAttack, and CW across multiple target labels. While the current version lacks these additional attacks, we believe the presented experiments are sufficient to ground our theoretical contribution, which is the primary focus of this paper, and respectfully request that the reviewer considers these extensions for future work.
Please find the updated manuscript attached. Changes to the reference section are highlighted.
Best regards,
The authors

Reviewer 2 Report
Comments and Suggestions for Authors
Dear authors,
thanks for such an easy piece to review! I quite liked your presentation of the issue and the reference to the extensive appendices were really informative.
I also believe your results show the IB should be modified for supervised tasks, as my experience also leans towards poor results on classification when representation and classification are trained separately.
My only peeve is that the references often lack page numbers and other details, and there are a number of links broken to the original sources. I am attaching my anonymously reviewed version of your paper and you can actually concentrate on just the references to try and find those with which I found fault.
Regards,

Author Response
Dear Reviewer,
Thank you for your valuable and encouraging feedback. We're very pleased that you found the presentation clear, and the methodology sufficient to show that the IB should indeed be adapted for supervised tasks.
Thank you for pointing out the discrepencies in the refrences section, and for providing the detailed pdf with the required corrections. We've revised the bibliography to fully align with Entropy’s formatting guidelines (link), added missing details such as page numbers, conference locations, and dates where available, and removed or corrected broken links.
Please find the updated manuscript attached, with changes to the references section highlighted.
Best regards,
The authors

Round 2
Reviewer 1 Report
Comments and Suggestions for Authors
The author solves all my concerns, and I would like to recommend accepting the paper.